# Violence and Corruption of Megachurch Leaders: Unravelling Silent Coloniality in Zimbabwe

**Bekithemba Dube**

Faculty of Education, University of the Free State, Bloemfontein 9300, South Africa; dubeb@ufs.ac.za

**Abstract:** This theoretical article argues that megachurches are an inadequately problematised factor in the Zimbabwean crisis and uses, as examples of violent and corrupt megachurch leaders, Emmanuel Makandiwa, Uebert Angel, and Passion Java. As Zimbabwe moves towards elections in 2023, ZANU-PF has resorted to using megachurches to enact propaganda, create voter empathy, and stir up violence, dividing the religious electorate along party lines in the process. The article is couched in decoloniality theory to position megachurch leaders within instability and as thwarting democracy in Zimbabwe. I respond to two questions: how do Makandiwa, Angel, and Java contribute to thwarting democracy while promoting corruption and violence? And, how can religion be approached from the perspective of decolonial thinking to reverse the crisis that has been created by prophets in Zimbabwe? I end by arguing that the Zimbabwean crisis takes various forms and that the role of megachurch leaders in finding a solution and in reconstructing narratives of peace and good governance in Zimbabwe cannot be ignored.

**Keywords:** Passion Java; Emmanuel Makandiwa; Eubert Angel; religion; politics; Zimbabwe

## 1. Introduction

Much research has been conducted on the intersection of megachurches and politics in Zimbabwe and beyond. Megachurches are religious groupings that have large followings wherein a leader occupies a crucial position in relation to divine deity and is often equated to Christ (Banda 2018). Megachurches use language "embedded [with] logic that enforces control, domination, and exploitation disguised in the language of salvation, progress, modernization, and being good for everyone" (Mignolo 2005, p. 6). Often, megachurch leaders are commonly referred to as prophets (a term which has come to be associated with individuals believed to be connected to God and who speak directly with God, creating a degree of superhumanity). In most African countries, they use their influence for various purposes, chief among which is to neutralise people's anger about incompetent governments, especially when there are material benefits from the state involved. To cement the neutralisation, megachurch leaders resort to quoting biblical scriptures, such Romans 13, to convince people that all governments are God-ordained and, as such, that the duty of people is to support them. However, there has been a deliberate attempt to isolate Romans 13 from other biblical narratives such as Proverbs 28:12, which seems to give another version of how people should interact with politics in the context of repression. In fact, these two biblical passages have not been fully explored to tease a position in the context of both democracy and repression. Contextualising the article within the Zimbabwean milieu, Machakanja (2010) argues that people are "trapped in a complex and protracted political crisis that has seen rising levels of human-rights violations, including kidnappings, disappearances, arbitrary detentions, torture, sexual violence and the forced recruitment of youth by armed groups, to name just a few" (p. 2). Premised in the foregoing observation, some megachurch leaders in Zimbabwe are mute and often fail to confront and address these violations; instead, they support a regime that is responsible for human atrocities. Cognisant of this, megachurches enable crises and compromise the value of megachurches for reconstructing a democratic society.

Megachurches enact, among their followers, a mentality that leaders cannot be questioned or held accountable by people—only by God. Consequently, their views constitute the alpha and omega; hence, politicians see megachurches as political capital bases. The problem begins with the observation by Maldonado-Torres (2017) that "religion [megachurches] is seen as a category that does not deserve critical scrutiny, because there is a sense that, at the end of the day, we all know what religion is" (p. 691). Furthermore, the problem is exacerbated, as explained by Johnson (2015, p. 109), because "religion tends to be easily relegated exclusively to the realm of ideology or belief. This rendering obscures the overwhelming worldliness of religion and its profound role as a network of social institutions, materialises, technologies, and cultural practices". However, megachurches have a "political theology [which] deals with nation state, since there is no theology that does not deal with the nation state' (Ngong 2020, p. 2), though the dealing should be towards making the nation state better site for its citizens and ultimately contributing to democracy.

To develop the foregoing argument, I focus on three megachurch leaders in Zimbabwe, namely Emmanuel Makandiwa, Passion Java, and Uebert Angel. Much has been written about these three and, of course, from different perspectives, ranging from praise and listing their miracles to controversies. However, little has been done to contextualise these megachurch leaders in relation to their thwarting of democracy through their close relationship with ZANU-PF. Most of the information about these three leaders regarding their recent involvement in politics was gleaned from electronic news media and confirmed by checking other electronic platforms, such as YouTube; furthermore, some academic sources have written on them from different perspectives than those pursued by this article. The positive aspect of recent and electronic media is that fresh news continuously emerges about these megachurch leaders, as opposed to ideas about their lives and participation in Zimbabwean politics being recycled.

The three prophets are well respected by people who subscribe to the ideology of megachurches and by politicians aligned with ZANU-PF. This article should not be construed to imply that the three prophets have not contributed positively to Zimbabwean society; they have achieved much, but this article takes a different angle and presents a side that has been overlooked and, ultimately, contributes to the crisis. This side comprises their silence on repression, corruption, and the violation of human rights by ZANU-PF, which is of great concern. This silence by influential people triggered Magaisa (2019) to ask,

> why, in some circumstances, does evidence of mendacity, crudeness, or cruelty serve not as a fatal disadvantage, but as an allure, attracting ardent followers? Why do otherwise proud and self-respecting people submit to the sheer effrontery of the tyrant?

My intention is not to undermine the three prophets' constitutional right to belong to political parties of their choice; however, my bone of contention relates to respected megachurch leaders being mute about corruption, violence, and the abuse of citizens. Why do they not use their influence to help create a democratic and non-violent society? As will be shown, in some contexts, megachurches spearhead violence and corruption and those opposed to them are removed from the zone of being (Fanon 2008). This has led to Magaisa (2020) asking a critical question:

> we wonder why certain members of the clergy seem to take the side of an incompetent and corrupt regime. Does the regime have something on them, people ask?

In response to a failure to locate the logic behind why megachurches fall victim to ZANU-PF, it can be argued that they have embraced the philosophy that Ndlovu-Gatsheni (2009, p. 358) refers to as Mugabeism. In his description of the phenomenon, he argues that

> Mugabeism has never been a democratic phenomenon. It has been intolerant, violent, and ever ready to discipline any form of dissent. Its strategy of 'making the nation-as-people' has always been dominated by coercion, where autonomous trade unions,

*women's organizations, and even religious groups had to be subordinated to the nationalist imperative.*

The three prophets, as will be shown, operate within the parameters of enhancing Mugabeism by portraying ZANU-PF as the only God-ordained party to rule Zimbabwe. In this way, the prophets consolidate "ZANU-PF claim to be the alpha and omega rulers of Zimbabwe' (Ndlovu-Gatsheni 2009, p. 3), which, of course, is not a problem as long as the rights of all Zimbabwean people are valued—including the rights of those with dissenting voices.

Cognisant of the thwarted democratic space in Zimbabwe, which has been cemented by megachurch leaders, I concur with Maldonado-Torres (2017, p. 710), who believes in the need to question and problematise the binaries that religion poses today. In so doing, I believe decoloniality theory positions me to critique and argue for the democratic involvement of megachurches in politics, with their involvement being premised on love, care, and accountability, as an alternative for a better Zimbabwe.

## 2. Theoretical Framework: Decoloniality

To enact politics of love, care, and accountability, decoloniality theory is ideal to question, challenge, and problematise elements that thwart democracy. Decoloniality theory is "traceable to those thinkers from the zones that experienced the negative aspects of modernity such as Aime Cesaire, Frantz Fanon, William EB Dubois, Kwame Nkrumah, and Ngugi wa Thiong'o" (Ndlovu-Gatsheni 2015a, p. 490). I consider decoloniality to be relevant for this study since it allows a re-emerging within the context of a crisis of imagination relating to liberation, freedom, development, and the future (Ndlovu-Gatsheni 2015b, p. 22), which is pertinent for redirecting megachurches towards peace and stability. Decoloniality is used, in this article, to argue for the need to "dismantle power relations and conceptions of knowledge that foment the reproduction of racial, gender, and geopolitical hierarchies that came into being or found new and more powerful forms of expression in the modern/colonial world" (Grosfoguel 2011, p. 1). Decoloniality is ideal for this study since it involves a search for "better ways of theorising and explaining the meaning of religious liberation and freedom, as well as taking the struggles forward in contemporary surmising" (Ndlovu-Gatsheni 2015b, p. 23). Central to achieving this goal in the Zimbabwean space is people's ability to question why unequal power persists in sustained colonial relations through areas of being, knowledge, and subjectivity (Dei 2019). Kessi et al. (2020) see decoloniality as a theory that entails a political and normative ethic and as the practice of resistance and intentional undoing, that is, both unlearning and dismantling unjust practices, assumptions, and institutions as well as maintaining persistent positive action to create and build alternatives (Kessi et al. 2020, p. 271). The new alternatives framed in decoloniality will, in Mignolo's view, expose and challenge the "current control of the planet by the bourgeoisie [megachurch leaders], [which] has generated all kinds of conflict, discontent, humiliation, anger and dehumanisation" (Mignolo 2017, p. 44). By using decoloniality theory in this paper, I seek to reverse the foregoing and, to the contrary, champion for a convivial community, which is explained by Nyamnjoh (2017, p. 5) as being characterised by "diversity, tolerance, trust, equality, inclusiveness, cohabitation, co-existence, mutual accommodation, interaction, interdependence, getting along, generosity, hospitality, congeniality, festivity, civility and privileging peace over conflict, among other forms of sociality". Thus, by imagining the dairies of the future, decoloniality argues that the nexus between megachurches and politics in Zimbabwe should result in a convivial community regardless of which party is in power. In the following section, I will discuss the three prophets in detail, with a focus on instability and on the ways in which they thwart democracy in Zimbabwe.

### 3. Role of Makandiwa in Thwarting Democracy

Emmanuel Makandiwa is one of the most influential religious leaders in Zimbabwe. He is also known as Shingirai Chirume and was born on 25 December 1977. He grew up in Muzarabani, Mashonaland Central Province, where his family farmed cotton, grain, and other crops (Henotace.org 2019). He received his religious education at Living Waters Bible School, a theological institute of the Apostolic Faith Mission. He was a pastor at an AFM branch at How Mine, located about 30 km south of Bulawayo, after which he moved to the Chitungwiza branch near Harare, the capital city of Zimbabwe. When he was in Chitungwiza, he became very popular because of his messages and the miracles he performed. While pastoring at AFM, he ran a separate, parallel religious ministry that was attended by people from different churches. He became a household name, especially in Harare. This caused a collision with AFM elders, who wanted him to choose between AFM and his interdenominational ministry; consequently, he formed his own ministry in 2009, which is now known as the United Family International Church (UFIC). In addition to operating the ministry, Makandiwa is a decorated businessman with interests in logistics, media, and other ventures. His net worth is estimated at USD 1 billion (Gambakwe Media 2021). His followers are extremely loyal, to the extent that they are prepared to execute violence on his behalf due to the value they assign to him as a man of God (Gunda and Machingura 2013, p. 23). Considering this background, he became a man of interest to ZANU-PF to push the regime-enabling agenda.

I refer to Makandiwa in this section for the following reasons: First, he has declared himself to be a ZANU-PF supporter, which, of course, is a constitutional right (Mujinga 2018); secondly, in the context of violence by ZANU-PF against dissenting voices, Makandiwa is mute on the atrocities committed by ZANU-PF; thirdly, his position regarding corruption happening in Zimbabwe is noteworthy. Megachurch leaders such as Makandiwa, at face value, appear very professional, but become compromised due to rewards associated with aligning oneself with the ruling party. According to Magaisa (2019), megachurch leaders "start from the periphery wearing the label of technocrats but soon enough, they will find themselves deep in the cesspool, wearing scarfs and chanting ridiculous slogans". Megachurch leaders, like any other human beings, become trapped by the ruling party, by design or by default, and, in the end, as has been the case with Makandiwa, become compromised when it comes to corruption. To illustrate the foregoing, Makandiwa appeared to support corruption indirectly by arguing that

> Corruption has certain different levels. It has different levels. There are certain dimensions that if there is 10 billion. And the 10 billion is set aside to construct roads. If 5 billion is swindled and the other 5 constructs the road. Corruption is corruption, but this corruption is different from corruption because the whole 10 billion could have been swindled (quoted by Ndoro 2022).

Makandiwa believes corruption can be categorised; however, it is not clear whether he condemns even the lowest level of corruption. His views on corruption are contrary to those of the Zimbabwe Catholic Bishops Conference (ZCBC), who argue that "corruption in the Zimbabwe has reached alarming levels ... there hasn't been equally a serious demonstration by government to rid the country of this scourge" (ZCBC 2020). Given these scenarios, those who benefit from corruption tend to be mute about it, in this case, Makandiwa. Therefore, Togarasei and Biri (2018) are right to call megachurch leaders moneymaking machines that do not hesitate to use corruption to achieve this goal.

To conclude this point, corruption has never been good for any country; it is a form of repression against people whose resources are stolen by the powerful in partnership with megachurch leaders. Hence, decoloniality theory problematises corruption promoted by megachurch leaders such as Makandiwa as corruption causes instability that affects church members and other members of the general populace.

The second issue regarding Makandiwa, instability and the thwarting of democracy, is that he has militant followers who are prepared to fight for him, starting with those on

social media such as Facebook and Twitter (Gunda and Machingura 2013). Makandiwa's followers regard him as a very important man who should not be subjected to scrutiny since he was chosen by God to lead this generation. In a sense, Makandiwa's followers promote him as though he is God or someone replacing Christ (Banda 2018). To expand on this point, Makandiwa appeared in a video where he states that he is "more gifted than God" (ZimEye 2023), which, rightfully, stirred up controversy and debate. Through this, Makandiwa elevated himself more than God, which has had consequences and influences what he says to his followers. He can feed negatively on the politics of a country like Zimbabwe, since he aligns with a regime of repression, and cannot be questioned since he is "more gifted than God".

The scope of this article, however, is not to argue about whether Makandiwa is more gifted than God but to expose the violent responses of Makandiwa's followers, who were convinced that the man of the cloth was right to make this claim. One Facebook user, who goes by the name Taffy Theman, castigated Makandiwa for alleging that he is more gifted than God; his argument was that God is above all humans and that there is no category whereby a man can be compared to God. However, by problematising Makandiwa, Makandiwa's followers threatened Taffy Theman because Makandiwa may not be questioned regardless of what he says. This attitude can be referred to as mutual zombification (Mbembe 2002).

This mentality—of elevating religious leaders as being more than human—has never produced good results. We should remember the mass murder–suicide of Jim Jones' People's Temple in Guyana, where 918 people died of poisoning because they failed to resist and question their religious leader (Sanua 2007, p. 332). Similarly, we should remember Ervil LeBaron, who had 51 children by 13 wives and, over two decades, amassed hundreds of followers who allegedly murdered more than 20 people on behalf of LeBaron and on his orders (Levitan et al. 2021). Efforts to ignore the influence of people like Makandiwa often generate into crises as people hold on to the vision of the prophet, which can be compromised to thwart democracy.

## 4. Role of Uebert Angel in Violence

Uebert Mudzanire started using Angel as his surname after claiming to have encountered an angel. He is another of the popular prophets to have emerged in both the religious and political landscapes in Zimbabwe. He is currently Zimbabwe's ambassador-at-large for Europe and the Americas. He founded Spirit Embassy (later renamed Goodnews Church) in the United Kingdom and has expanded his religious services to an international scale (Chester-Londt 2022). In addition to religion, he is involved in various business ventures; owns companies such as Billion Group Ltd., the Millionaire Academy, Sam Barkley Construction, and Black Stallion; and has a net worth of USD 60 million (Nkosi 2022). *The Standard* describes Prophet Angel as follows:

> *Flamboyant and sharp tongued, Prophet Angel made his way into the public glare in 2011, championing the gospel of prosperity. The tall and lanky prophet, who dressed with a flourish, quickly became one of the most sought after prophets locally and regionally. Fondly referred to as 'Papa' or 'Major' by his congregants at Spirit Embassy, the youthful prophet—whose collection of flashy cars endeared him to the youthful group in his church—was unapologetic about his boastful tirades. He reached an all-time high with his miracle money sermons which almost got him into trouble with the Reserve Bank of Zimbabwe officials, but [he] wriggled his way out of it and became even more popular (Standard 2015).*

I discuss him in this article mainly because of his alleged involvement in illicit gold dealing, which was exposed by an Al Jazeera documentary about gold mafias in Zimbabwe (Al Jazeera 2023). In the documentary, Angel is seen being involved in parallel-buying and selling gold and escaping the consequences due to his diplomatic immunity. In the documentary, Angel is cited by Al Jazeera (2023) as saying,

*You want gold, gold we can do it right now, we can make the call right now, and it's done . . . It will land in Zimbabwe—Zimbabwe can't touch it too until I get to my house. So, there can be a diplomatic plan.*

The diplomatic plan he mentions was that he would not be searched at the point of entry and could, therefore, smuggle in money and gold without being challenged. This indicates that corruption also is rampant among men of the cloth, thereby affecting the economy and causing instability in the country. While the documentary would have attracted investigation, suspension, and even imprisonment if he was a citizen of a country where accountability is key for national development, prophets associated with ZANU-PF are immune from prosecution; as long you are a regime-enabler in Zimbabwe, you are exempt from judiciary processes that deal with social delinquency. An interesting consequence of this scenario was that instead of apologising to the Zimbabwean people, the prophet/ambassador threatened his critics, who reported on this, with violence. An unnamed journalist working at Bulawayo24 News (2023) quotes Ambassador Mudzanire as saying,

*Without praying, Phineas took a knife and stabbed the men he saw committing adultery, Munhu anongoda kurohwa mbama, zvekuti anyorereyi ichi, zvekunamata tombosiya! [Someone must be slapped for writing about this, let's put aside prayer at the moment].*

It is clear that Angel does not hesitate to threaten violence to silence his critics. This also implies that he believes that prophets should not be subjected to accountability, since critics will be met with violence. Angel is very clear that the Bible should be set aside to deal with certain issues, including criticism of his behaviour; the only option is to use violence. This locates megachurch leaders among the originators and sustainers of the Zimbabwean crisis. Furthermore, there is an implication that megachurches are safe havens for violent people who, when they commit crimes, become active agents of ZANU-PF and are prepared to unleash violence on behalf of the state, not as a way of defending it but to get protection. Using decoloniality thinking, I problematised the foregoing premised on the thinking of Ndlovu-Gatsheni (2013, p. 6), who stated that

*what Africans must be vigilant against is the trap of ending up normalizing and universalizing coloniality as a natural state of the world. It must be unmasked, resisted and destroyed because it produced a world order that can only be sustained through a combination of violence, deceit, hypocrisy and lies.*

To redirect megachurches towards supporting development in Zimbabwe, megachurch leaders' authority should be questioned. Failure to do so, according to Dreyer (2007), means "religion can to be exploited in power struggles by those in power, or those seeking power". In the following section, I will discuss Passion Java.

## 5. The Role of Java in Violence

The Zimbabwean megachurch landscape has never been without drama and surprises, especially since the advent of Panganai Passion Java in both the prophetic and political spaces in Zimbabwe. He doubles up his prophetic role with comedy, and through the comedy, he emerges as a political and religious nuisance, which, I argue, serves to consolidate the crisis in Zimbabwe.

Passion Java was born on 28 October 1988 and is the founder of the Passion Java Ministries and the Kingdom Embassy church (Pindula News 2019). According Godfrey Kurauone (reported by Mpofu 2022), Passion Java is a self-proclaimed prophet who enjoys permanent resident status in the USA and periodically returns to Zimbabwe to actively support a regime that is killing citizens, violating fundamental rights, and stealing national resources. He is very clear that he supports ZANU-PF, viewing it as the only relevant party to lead Zimbabwe as, he claims, ordained by God. In his role as a comedian who jokes about repression, he commands a huge following, especially among the youth of Harare, who see him as a model of wealth and empowerment as a result of his association with ZANU-PF. AfricanTimes (2021) reports on claims that "Mnangagwa is recruiting

Zimbabwean social media celebrities and well-known men of God in a bid to acquire many votes for the upcoming 2023 election". Thus, ZANU-PF members see him as an important figure who can lure the youth into supporting ZANU-PF and often refer to him for evidence of successful policies that the government has implemented to empower citizens. This is in line with the observation by Gramsci (1999, p. 258) that

*Every State is ethical in as much as one of its most important functions is to raise the great mass of the population to a particular cultural and moral level, a level (or type) which corresponds to the needs of the productive forces for development.*

While each state may have good intentions for its citizens, the praxis of these intentions often becomes very difficult in the context of suppressing dissenting voices. For ZANU-PF, Passion Java is a critical element when it comes to muting dissenting voices, especially among the youth. He flashes the riches he has accumulated in the USA and claims he earned them through empowerment by ZANU-PF. In doing so, he operates within the premise of Brittain (2012, p. 206), who argues that religious leaders champion "ideological support of the nation-state, encouraging people to passively bear injustice". Passion Java encourages youths to join ZANU-PF by promising them that, perhaps, they can make it in life like he did since ZANU-PF provides such a platform. His strength, which gives ZANU-PF the edge, is that he uses soft violence—jokes—as a form of repression. Serguei Alex Oushakine (2011, p. 657) comments on the use of jokes as repression:

*they are not the only genre that channelled comic content during socialism. Yet, unlike many other forms of publicly circulated laughter, they expressed in an unusually salient way how the regime we loved to hate became an object of our affective—albeit negative—attachment: A political intimacy with a crooked smile, achieved through mocking and ridicule.*

Through these jokes, Java resorts to shaming ZANU-PF opponents on various social media platforms. It is in this context that I problematise Java as a prophet who is violent and contributes to the Zimbabwean crisis. Thus, people that resent ZANU-PF are silenced by Java through jokes of repression. Considering this, the ZCBC warns that "the suppression of people's anger can only serve to deepen the crisis and take the nation into deeper crisis, the crackdown on dissent is unprecedented" (ZCBC 2020).

I problematise Passion Java largely because of the way he body-shames people in an attempt to silence his critics. He referred to businessman Genius Kadungure, aka Ginimbi, as having a 'handsome' face and a 'fist-like' nose (Makuwe 2020). Java called Hopewell Chin'ono, a Zimbabwean journalist who was jailed for exposing COVID-19 corruption by ZANU-PF bigwigs, '113010', a street name used to refer someone as a penis (ZimboCelebs 2021). This was after Chin'ono said that youths who followed Passion Java were not focused since Java was just a comedian and that following him would not result in solving the crisis in Zimbabwe. About Talent Chiwenga, a street evangelist, Java said he had a face like a foreskin and that his surname was Chiweti, meaning he was urine. This was after Chiwenga referred to Passion Java as a disgrace and as playful (ZimEye 2021). The list of people he has ridiculed is endless; he does it in a comedy format, which keeps youths entertained but damages the reputations of the targeted political and religious actors who oppose him and ZANU-PF. Through these body shaming utterances, he silences people by ridiculing them and violating their rights. He uses jokes of repression "to legitimise, sustain and even promote political tyranny and oppression" (Zimunya and Gwara 2013, p. 188). In short, Passion Java, as a megachurch leader, contributes to the social crisis and to the negative perceptions of megachurches and politics in postcolonial Zimbabwe. Hence, there is a need for "groups, communities, societies, and organizations to promote social change" (Given 2008, p. 140) by challenging oppression where it appears since it does not contribute to promoting a democratic space.

To promote social change, in the following section, I will make concluding remarks on religious leaders and the way they could interact with the state to ensure that religion

contributes to democracy, social transformation, and the appreciation of different people regardless of political orientation.

## 6. Decolonial Thinking on Religion in Post-Colonial Zimbabwe

Given the context described in the previous section, it is inevitable that we have to think anew about megachurches, with a new lens, to rehumanise and move all Zimbabweans to the zone of being. The decoloniality approach offers much promise. While problematising the three prophets does not bring immediate change in the Zimbabwean political system, it teases and serves as a reminder that scholars prone to social justice should continually unearth and challenge social injustice wherever it appears, even in a religious context. This is premised on the view that decoloniality has been a systematic project of removal; a pedagogy of and towards decoloniality must also be a project of "re"—resisting, refusing, rehumanising, remembering, reminding, restoring, reframing, revisioning, and reimagining (Reyers 2019, p. 7). It is clear from the actions of the three prophets that the exclusionary approach of the three prophets constitutes a deliberate attempt to thwart dissenting voices against ZANU-PF, which weakens any effort to enact a better Zimbabwe as a democratic space. Thus, decolonial theory is ideal to emancipate the excluded and subordinated subjectivities and to search for a new base from which to launch a new world order that is humane and inclusive (Ndlovu-Gatsheni 2015b, p. 22). The possibility of this happening relies on the people who are oppressed by religion to expose, resist, and challenge the abuse of religion in the political space. This does not suggest that decoloniality calls for delinquency. To the contrary, it calls for both civil and epistemic disobedience, which could be enacted at different levels and in different spheres (Mignolo 2017, p. 41) to redress the religious abuse that has created a crisis in Zimbabwe. I argue that the recovery of Zimbabwe in all spheres of life requires a new rhythm that is specific to a new generation of people with a new language and a new humanity (Fanon 2004, p. 2). In doing so, as Tlostanova and Mignolo (2009) contend, people are removed from the zone of nonbeing to transmodernity, which is characterised by the epistemic and ontological dwelling of "the other", who has been rendered to the periphery via asymmetric power arrangements—in this case, by ZANU-PF and the prophets. I also submit that the crisis in Zimbabwe requires the "megachurches to be agent of transformation that seeks to confront, interrogate and engage prophetically powers that be, including issues of racism, forgiveness and reconciliation" (Mashau 2018, p. 6). Regarding Emmanuel Makandiwa, Uebert Angel, and Passion Java, a better Zimbabwe will be good not only for those suffering but also for the perpetrators of injustice since decoloniality is a not a revenge journey but an invitation for both the perpetrators and the victims to walk together along the journey of the rehumanisation of the dehumanised—a journey that provokes the courage to care and to love (Mpofu 2017) despite political and religious disparity.

In short, the attempt to problematise the three prophets was born from a realisation that megachurch leaders have not been well problematised as contributors to the crisis in Zimbabwe; hence, this investigation, using decolonial thinking, forms part of the struggle to "regain lost subjecthood and eventually citizenship, as well as many other questions to do with being and humanism as politicised states of existence" (Ndlovu-Gatsheni 2015a, p. 491). Of course, this does not mean that prophets cannot champion their political parties of choice; however, as they do so, they should side with all people, regardless, especially with those who are poor and marginalised and those who have been dehumanised and suffer because of political arrangements. In short, Zimbabwe is in crisis, and any attempt to address it should take cognisance of megachurches, which have not been adequately problematised as a factor in Zimbabwean politics. Hence, megachurch leaders should unmute when injustice is evident, since failure to do so creates crises and instability.

## 7. Conclusions

In this article, I exposed how Emmanuel Makandiwa, Uebert Angel, and Passion Java have contributed to crises in Zimbabwe. I showed how their nexus with ZANU-PF has created instability and skewed nationalism. Decoloniality invites both those who have been dehumanised as well as the perpetrators of crises to engage in a journey to restore a better Zimbabwe where religion contributes to sustainable development. I conclude that prophets play a key role in Zimbabwe and that their influence should be used to reinvent a better Zimbabwe in which accountability is key and respect for human rights is emphasised for all, regardless of political and religious orientations.

**Funding:** This research received no external funding.

**Institutional Review Board Statement:** This study was ethical cleared by the University of the Free State, UFS-HSD2022/1559/22.

**Conflicts of Interest:** The author declares no conflict of interest.

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
