# Peer review of "Violence and Corruption of Megachurch Leaders: Unravelling Silent Coloniality in Zimbabwe"

_religions, doi:10.3390/rel14091209_

Round 1
Reviewer 1 Report
1. The article attempted to evaluate three prophets from Zimbabwe from a perspective of decoloniality. The article argues that Makandiwa, Angel and Java are agents of coloniality and promote violence and corruption in Zimbabwe.
2. While I find the concerns raised by the article concerning the violence and corruption promoted by the named prophets to be great importance, I find the article to be lacking in depth and scholarship. I find the article to be inadequately researched and the much of the evidence presented about the prophets is drawn from questionable sources, such as Zimeye, a discredited online news site. It seems the article only limited itself to scanty facts drawn from unsubstantiated social media gossip. For the article’s claims about the selected prophets to stand more in-depth and critical research needs to be done. There is a considerable volume of scholarship on Emmanuel Makandiwa, Hurbert Angel and Passion Java for the article to construct a fair and just critical assessment.
3. It is not clear how the theme of coloniality is applied to the named prophets, for the theme of coloniality to not function as window dresser it is important to show clearly the aspects of coloniality among prophets.
4. I could also argue that coloniality is probably not the best theoretical framework of analysing these prophets.
5. There are several instances of language mistakes that need to be attended.
6. I recommend that the article be thoroughly revised.

Author Response
Dear Reviewer
Thank you for the insightful reviews. I appreciate and i learnt a lot. I hope the article has improved. I am open for further suggestion. in the meantime thank you so much

Reviewer 2 Report
This is an interesting account of an unfortunately common phenomenon across Africa, but specifically in terms of the Zimbabwean context: the legitimation of "strong man" regimes by charismatic leaders of megachurches. These leaders, "strong men" in their own right, serve as an extension of the ruling party into religious institutions. This has the effect of silencing legitimate dissent, indeed of representing that dissent as "satanic" or "antichrist," and of suppressing the agency of "the suffering masses." This suppression is, I think, where the interest in "decoloniality" comes in.
The article is fairly clear in its structure: the problem is stated; the proposed analytical framework is suggested; three examples are given; then the analysis is applied.
There are, however, some problems with the article that need to be addressed before publication: the conceptualisation of "religion" is unclear; the tone is overly rhetorical; and the relation of the analytical framework to the examples needs more explicit working out.
First, the use of the term "religion." The term itself is not without problems. It has a peculiar genealogy within European modernity (the classic work of Wilfred Cantell Smith is still useful), and is ambiguous in colonial, African contexts, as David Chidester and others have shown. The way it is used in the article is confusing. Sometimes it is used as a general phenomenon, as in "religion is an inadequately problematized _factor_ in the Zimbabwean crisis." (line 5, emph mine) Other times it is identified with particular "religious leaders" the author finds abhorrent (e.g. line 24), or "[other] religious actors opposed" to these particular religious leaders (e.g. line 345). Here the term "the religious sector" (line 96--which includes the three under discussion but also others, e.g. the Catholic Bishops--line 333) is a more encompassing term for such leaders, inclusive of both good and bad in the author’s view. Sometimes "religion" seems to have its own agency (e.g. line 38) but other times it is "used" by other agents "to legitimate… tyranny". (e.g. line 347). The term is also used to refer to the author's own (?) "Christian religious perspective" (e.g. line 351). There is also the question of African religions displaced or transformed by colonial Christianity. This is not taken into account. Indeed, David Ngong has shown that the “strong man” figure may well be pre-colonial and pre-Christian.
Some clarification here would help.
My suggestion would be for the author to avoid the term "religion" where possible. This would mean talking about "certain megachurch leaders" instead of “religion.” Doing so would also allow comparison with similar leaders in other places on the continent, as well as the growing literature on them. It would also help clarify the use of the term "prophet" of these three, which can be a technical term within religious studies or a self-appointed designation.
The second problem concerns the tone of the article: is it scholarly description, the use of a specific frame to give insight into a problem that has puzzled others? The advocacy of a particular strategy bring liberation to oppressed people? The rescue of religion from those who would hold it hostage to ideological interests?
While these are not mutually exclusive, they should be distinguished, and more rhetorical or “preachy” expressions (e.g on line 51: "as long as you are a regime enabler in Zimbabwe, you are exempt…") curtailed. I also wonder how the author's own "Christian" perspective connects to the coloniality of Christianity (see line 24) as part of the problem identified in decolonial thought.
The third observation follows from this, and concerns the theoretical framework used. "Decoloniality theory" (line 113) is talked about both as a descriptive, analytical lens, a fresh way of seeing a problem. But the way it is actually employed in the rest of the article seems more appropriate to prescribing an activist agenda: the recovery of the agency of the poor. I may have missed it, but it's not clear how "coloniality" provides a new _description_ of the situation in Zimbabwe as concerns religious leaders. Indeed, the first line of the article says that “[m]uch research has been done on the intersection of religion and politics in Zimbabwe” and that “religion has always been used in the political space,” but no references or examples of such research are given. Much of what the article states about the religious and political corrupt leaders could be said without talking about "coloniality." At the very least the author could spell out how coloniality compares to other ways of analyzing the use of religion in Zimbabwean politics.
I hope the author can take these things into account and resubmit the article.
References:
Chidester, David. Religions of South Africa. London: Routledge, 1992. Print.
Ngong, David T. “Recent Developments in African Political Theology.” Religion Compass 14.10 (2020): 1–11.
Smith, Wilfred Cantwell. The Meaning and End of Religion. Minneapolis: Fortress Press, 1991.
A few idiosyncratic expressions a good editor should be able to correct fairly quickly.
Author Response
Dear Reviewer
Thank you so much for your suggestions and indeed they have improved the quality of the paper. I tried my level best and I am sure you will notice improvement. I am open for further suggestions to make the article better
Thank you again

Round 2
Reviewer 1 Report
The article has greatly improved, and I recommend that it be published.